# The Role of Neutrophils in Multiple Sclerosis and Ischemic Stroke

**DOI:** 10.3390/brainsci14050423

**Published:** 2024-04-25

**Authors:** Anna Nowaczewska-Kuchta, Dominika Ksiazek-Winiarek, Piotr Szpakowski, Andrzej Glabinski

**Affiliations:** Department of Neurology and Stroke, Medical University of Lodz, ul. Zeromskiego 113, 90-549 Lodz, Poland; anna.nowaczewska-kuchta@umed.lodz.pl (A.N.-K.); dominika.ksiazek@umed.lodz.pl (D.K.-W.); piotr.szpakowski@umed.lodz.pl (P.S.)

**Keywords:** neutrophil, multiple sclerosis, ischemic stroke, central nervous system, blood–brain barrier

## Abstract

Inflammation plays an important role in numerous central nervous system (CNS) disorders. Its role is ambiguous—it can induce detrimental effects, as well as repair and recovery. In response to injury or infection, resident CNS cells secrete numerous factors that alter blood–brain barrier (BBB) function and recruit immune cells into the brain, like neutrophils. Their role in the pathophysiology of CNS diseases, like multiple sclerosis (MS) and stroke, is highly recognized. Neutrophils alter BBB permeability and attract other immune cells into the CNS. Previously, neutrophils were considered a homogenous population. Nowadays, it is known that various subtypes of these cells exist, which reveal proinflammatory or immunosuppressive functions. The primary goal of this review was to discuss the current knowledge regarding the important role of neutrophils in MS and stroke development and progression. As the pathogenesis of these two disorders is completely different, it gives the opportunity to get insight into diverse mechanisms of neutrophil involvement in brain pathology. Our understanding of the role of neutrophils in CNS diseases is still evolving as new aspects of their activity are being unraveled. Neutrophil plasticity adds another level to their functional complexity and their importance for CNS pathophysiology.

## 1. Introduction

Inflammatory response in the central nervous system (CNS) is defined as neuroinflammation. This process can have positive and negative aspects, which depend on the duration and the environmental context. Neuroinflammation involves various factors, like reactive oxygen species (ROS), adhesion molecules, cytokines, and chemokines, which are produced by endothelial cells, glia (microglia, astrocytes), and peripherally derived immune cells [1]. Pathological neuroinflammation is observed in primary damage caused by physical damage or the infection but also by vascular occlusion, cell death, or ischemia. Chronic high inflammation leads to neuronal damage and the development of neurodegenerative disorders. Negative aspects of neuroinflammation are associated with elevated levels of numerous proinflammatory factors, like ROS, IL-1, IL-6, TNF, IFN-γ, and CCL2 [1].

During CNS disorders’ development, like multiple sclerosis (MS) or stroke, non-immune and immune cells of cerebral tissue secrete chemoattractant mediators and induce expression of surface receptors that are responsible, among others, for leukocyte infiltration into the damaged region of the CNS [1]. This process further exacerbates blood–brain barrier (BBB) damage, increases proinflammatory cytokine milieu, and results in elevated neurological injury and cell death [2]. One of the first cells migrating from the periphery to the affected area of the brain is neutrophils. Neutrophils are cells of the immune system with short lifespans and are the most numerous group among granulocytes [3]. They are the body’s first line of defense against infections. Due to the presence of appropriate receptors on the surface of their cells, they can react quickly. Neutrophils are also able to initiate and maintain autoimmune neuroinflammation [2]. They can increase the BBB permeability by producing cytokines, ROS, and matrix metalloproteinases (MMPs) [4,5]. Neutrophils can act as alternative APCs but also can contribute to the maturation of microglia and infiltrating monocytes [6,7]. They also attract other immune cells to the side of inflammation. Moreover, diminishing neutrophil infiltration into the CNS after brain injuries limits neuronal damage [8]. This further supports their important role in neuroinflammation development. Neutrophils can be divided into different subpopulations based on the presence of surface receptors and secretory activity. There are subtypes with opposite functions, such as pro- or anti-inflammatory roles. Depending on the predominance of one of such subpopulations, there will be different functional effects. Skewing neutrophils toward the anti-inflammatory phenotype before stroke induction resulted in a reduced infarct volume [9]. The inflammatory processes may also alter the biology of neutrophils. Their life span may be increased due to the exposition to specific cytokines. Another subpopulation of neutrophils is called reverse-transmigrated neutrophils (rTEM). They have the ability to travel back to the bloodstream after infiltrating the injured tissue [10]. Elevated number of rTEM neutrophils was observed in patients with chronic inflammation [11].

The primary goal of this review is to discuss the current knowledge regarding the important role of neutrophils in the pathogenesis of two CNS diseases—multiple sclerosis and stroke as in both of them, neuroinflammation plays a central role in their progression, and neutrophils are one of the first cells present in the CNS at the onset. MS is an autoimmune disease. In contrast to this, stroke is a vascular disorder. Their pathogenesis is completely different, so it gives the opportunity to analyze the diverse mechanisms of neutrophils’ involvement in brain pathology. Despite various etiologies, there is evidence suggesting that there are similarities in the neutrophil importance for both diseases. In this review, we discuss the role of neutrophils as initiators of inflammation and as promoters of thrombosis. We believe that some aspects of these cells’ biology that are unraveled in one disease may be implemented in the second one. This is crucial because we observed an increasing number of young patients with MS and stroke. More studies in this field may facilitate a decrease in mortality and disability among these patients.

## 2. The Role of Neutrophils in Multiple Sclerosis

MS is a chronic, inflammatory, autoimmune demyelinating disease of the CNS [12]. MS has different clinical phenotypes: clinically isolated syndrome (CIS), relapsing-remitting MS (RRMS), primary-progressive MS (PPMS), and secondary progressive MS (SPMS) [12]. It depends on disease activity assessed by MRI and the progression of disability [12]. CIS describes the first clinical onset of potential MS and may be CIS of the optic nerves, brainstem, or spinal cord [13]. Moreover, this can be a dynamic process. One subtype may change over time in another [12]. MS mostly affects young people with a female predominant disease. Men are more likely to develop PPMS in all other MS phenotypes female predominant. MS males have a worse prognosis than females. Men have a higher rate of transitioning to SPMS [14]. Despite many studies, the exact etiology of MS remains unknown. Immunological, infectious, genetic, and environmental factors are considered to play an important role [14,15]. Recently, there have been many studies that indicate neutrophils’ possible involvement in the pathogenesis of multiple sclerosis [4,5,16,17,18,19].

Neutrophils have many ways to act and fight against microbes [5]. They are able to produce neutrophil extracellular traps (NET), release inflammatory mediators and enzymes such as IL-1β, MMPs, myeloperoxidase (MPO), produce ROS, produce and present antigens [5]. They can also destroy and phagocytize myelin and increase the BBB permeability [4,5]. 

### 2.1. NLR as Promising Indicator of MS

Numerous studies have highlighted the importance of the value of neutrophil-to-lymphocyte ratio (NLR) in MS patients. Researchers have studied various correlations related to NLR and MS. This rate was elevated in MS patients in general and in patients with optic neuritis [16,20,21], higher in MS patients with relapse [16,22], related to disease activity [23] and to the disability of MS patients [24]. Researchers also found that patients with elevated NLR had a higher risk of MS relapse within two years [22]. According to Bisgaard et al., NLR was weakly correlated with MS severity score (MSSS) [20]. In another study, the Expanded Disability Status Scale (EDSS) was correlated with higher NLR [25]. Moreover, according to Demirci et al., a high NLR was an independent and significant predictor of disability progression, which was estimated as EDSS [26]. In contrast to these studies, in another research, there was no association between NLR and EDSS, and also, no difference in NLR was found between progressive MS [16] and RRMS patients and neither between SPMS and PPMS patients [16]. Hemond et al. have shown that elevated NLR was associated with disability as assessed by the EDSS in MS patients independently of other demographic, clinical, treatment-related, and psychosocial factors [24]. In the context of annualized relapse rate (ARR), no association was found between ARR and NLR values [27]. Overall, these results suggested that this is a promising indicator worth paying attention to. In addition, obtaining this indicator is inexpensive and easy to carry out. Apart from being of significance as a prognostic marker in other diseases, such as stroke and autoimmune diseases, some studies also highlight the importance of MS. In the future, it may facilitate the diagnosis and early treatment of patients with MS.

The significant role of neutrophils in the onset of MS and during relapse was brought to the attention of Kostic et al., who detected neutrophils in the cerebrospinal fluid (CSF) during these disease stages. However, they observed a decrease in the number of neutrophils in CSF of patients with a long duration of the disease [27]. The role of neutrophils was also studied on an animal model of MS—an experimental autoimmune encephalomyelitis (EAE). Neutrophils were detected in the mice CSF during the hyperacute (transient) stage of EAE [28,29]. In other studies, as early as 24 h after EAE induction, neutrophils appeared in the meninges of mice, and their numbers increased during the preclinical and acute phases [17,18]. In another study, during the peak of EAE, much more neutrophil infiltration was found in spinal cord parenchyma [30]. As can be seen from these studies, neutrophils play a significant role during the initiation of MS disease and during relapse. As mentioned earlier, neutrophils have many ways to act and fight against microbes. Some of them may also influence MS development. 

### 2.2. Neutrophils Secretory Activity in MS

One such way is the production of NET, which is sometimes accompanied by a necrosis-like, programmed form of cell death called NETosis [31]. Neutrophils produce NETs to capture and kill microorganisms. Moreover, NETs also cause dendritic cell (DC) and T lymphocyte activation and are dependent on ROS and ATP production in humans [7]. It was found that the level of NETs is elevated in the serum of MS patients compared to the control group. It turned out that NETs were more abundant in subgroups of patients with RRMS and in men, whose prognosis is usually worse than in women [19,32]. Some results suggested that NETs may be toxic to the BBB and cause damage to adjacent neurons and other CNS cells [32]. Studies conducted on the EAE model showed that depletion of proteins (MPO, neutrophil elastase <NE>) associated with NET reduced disease severity and increased BBB integrity [33,34]. NE and MPO were also increased in the plasma of MS patients [28]. All of these studies indicated that neutrophil products (NET, MPO, NE) are elevated in MS patients and higher in remitting-relapsing MS (RRMS) patients and in men. The severity of MS is also related to the amount of NET-related proteins. 

Another mechanism by which neutrophils fight microbes is by affecting the IL-1 system, which is associated with neuroinflammation and repair in the CNS. IL-1 was observed in brain lesions of MS patients. Neutrophils contain ready-to-release IL-1β in their vesicles and may independently contribute to the maturation of IL-1β by secreting MMPs and serine proteases. Macrophages and DCs also produce IL-1β. However, neutrophils during EAE are the main source of this cytokine. It has been noted that the level of IL-1β increases in myeloid cells during EAE [35,36]. In IL-1R1 knockout mice, the onset of EAE was delayed, and the course of EAE was milder [37]. In addition, neutrophils and monocytes increase their IL-1β production in response to granulocyte-macrophage colony-stimulating factor (GM-CSF) [35]. GM-CSF in the CNS promotes the differentiation of monocytes into monocyte-derived macrophages and monocyte-derived DC. These cells can directly promote demyelination and axonal loss. In addition, GM-CSF can activate CNS-resident microglia, which produce proinflammatory mediators [38]. Also, neutrophil infiltration in EAE may be related to the expression of granulocyte colony-stimulating factor (G-CSF), as it correlates with the number of infiltrating cells. G-CSF administration was associated with exacerbation of EAE symptoms in a study by Rumble et al., and mice deficient in granulocyte colony-stimulating factor receptor (G-CSFR) were more resistant to EAE [28].

Another molecule secreted by neutrophils is the MPO enzyme, which converts hydrogen peroxide into reactive, secondary oxidants, which is a part of the “oxygen burst”. This creates a cytotoxic environment against pathogens [39]. Researchers Zhang et al. inhibited MPO during EAE and observed a reduction in EAE severity, reduction in absolute neutrophil count, and complete restoration of BBB integrity after a few days [33]. Recently, it was believed that phagocytosis of the myelin layer in MS was mainly carried out by macrophages and microglia. The possibility of neutrophil participation in this process has now been proposed [40]. Neutrophils are visualized in active MS lesions at sites where the BBB is destroyed. Researchers observed that neutrophils in EAE migrate through BBB more efficiently than neutrophils in the control group [41]. In addition, neutrophilic granules contain MMP-9, which is involved in BBB breakdown [42] and is elevated in RRMS and SPMS patients [43]. MMP-9 facilitates the infiltration of leukocytes into CNS, mediates T-cell migration through BBB, degenerates myelin sheath, and damages neutrons [43]. Smriti et al. found that mice with knock-out of MMP-9 and MMP-2 were resistant to the development of EAE [44]. 

Peripheral blood neutrophils obtained from MS patients are more primed, accompanied by an increased oxidative burst [19]. ROS production is directly involved in demyelination and axonal and astrocyte damage, both in EAE and MS [4]. ROS are also critical molecules for the activation of MMP enzymes [45]. According to the mentioned studies, neutrophil products such as IL-1β, MPO, ROS, and MMP [19,35,36] affect the severity of EAE, destroy the BBB, and their inhibition reduces the severity of EAE [32,33,34,37,44]. Neutrophils have many modes of action, and the dysfunction of even one of them may affect the course of EAE [32,33,34,37,44]. 

### 2.3. Role of PAD and CXCL1 in Demyelination and BBB Breakdown

Abnormalities of citrullination are known to play a pathogenic role in MS. Citrullination is the enzymatic conversion of arginine residues into citrulline, catalyzed by the enzyme pepridyl arginine deaminase (PAD) [46]. PAD4 is important for the NET formation and knockout of PAD4 on neutrophils, resulting in a neutrophil deficiency in the capacity to kill bacteria [47]. Resting neutrophils have PAD4 expressed on their surface and secrete PAD2. Neutrophil activation by some stimulatory factors increases the levels of PAD4 on the cell surface [48]. It was shown that PAD2 was increased in the MS brain. Moreover, research showed that inhibition of PAD attenuates EAE at all stages of disease progression. Both PAD2 and PAD4 are also localized in myelin and oligodendrocytes and take part in mechanisms of demyelination [49]. A novel mechanism of demyelination suggests that myelin breakdown takes place in the periaxonal region. In the previous hypothesis, myelin degeneration starts from the outside surface of myelin. The presence of PAD enzymes and myelin-associated proteases leads to the degeneration of myelin [50]. Production of PAD2 by neutrophils and expression of PAD4 on their surface may suggest the role of neutrophils in demyelination via this mechanism. 

The other factor attributed to neutrophils is CXCL1. This is a chemoattractant for neutrophils, which express the chemokine receptor CXCR2. In the preclinical phase of EAE, upregulation of CXCR2 expression on neutrophils was observed [51]. Jonathan et al., in their study, showed that persistent expression of CXCL1 in the CNS increased the severity of EAE and was associated with the recruitment of neutrophils to the spinal cord [52]. Another study showed higher neutrophil infiltration after CXCL1 administration in the preclinical phase of EAE [18]. Many studies point to the role of CXCL1 and the CXCR2 receptor during the effector phase of EAE [52,53]. Blockade or genetic silencing of CXCR2 in mice prevents BBB breakdown, reduces CNS leukocyte infiltration, and reduces the development of clinical deficits in EAE. In addition, injection of CXCR2+ polymorphonuclear leukocytes into CXCR2− mice restored susceptibility to EAE. Also, the depletion of these cells, including neutrophils, had a similar effect [53,54]. Neutrophils accumulate in the circulation prior to the clinical onset of EAE in response to systemic upregulation of G-CSF and the ELR+CXC chemokine CXCL1. G-CSF receptor deficiency and CXCL1 blockade inhibited the accumulation of myeloid cells in the blood and improved the clinical course in EAE mice [28]. 

### 2.4. Phenoplasticity of Neutrophils and MS

Neutrophils of healthy donors do not express major histocompatibility (MHC) class II antigens [55]. However, in autoimmune diseases, human and mice neutrophils may act as alternative antigen-presenting cells (APCs) for both T cells in the LN(lymph nodes) interfollicular region and B cells in the spleen and express MHC class II molecules [6]. Surface expression and intracellular MHC class II were observed when polymorphonuclear cells (PMN) had been cultured with IFN-γ and GM-CSF or with T cells or T cell-derived supernatants. PMN also acquired CD 80 and CD86 and increased expression of ICAM-1. Antibodies against MHC class II or CD 86 or ICAM-1 inhibited PMN-dependent T cell proliferation [54]. This is important because, in the pathogenesis of MS and EAE, a key step is the contact of T cells invading the CNS with related APCs and activation of autoreactive T cells [55]. Neutrophils can act as alternative APCs but also can contribute to the maturation of microglia and infiltrating monocytes [6,7]. Blockade of proinflammatory cytokines (IL-6, IL-12/23p40, IFN-γ, and TNF-α) produced by neutrophils showed only a slight reduction of MHC class II on bone marrow DCs [7]. Therefore, CNS-derived neutrophils, besides proinflammatory cytokines, provide currently unknown additional soluble proteins that mediate this process [7]. It was shown in vitro that neutrophil depletion caused impaired maturation of monocytes/macrophages and microglia into MHC class II expressing professional APCs [7]. 

Interestingly, neutrophils are not a homogeneous population, as was thought previously. There are various subpopulations of these cells. The classification of tumor-associated neutrophils (TAN) distinguishes two groups: N1-phenotype and N2-phenotype [56,57]. N1 neutrophils have anti-tumor activity, have pro-inflammatory role, highly express ICAM-1, produce proinflammatory cytokines (like GM-CSF, TNF-α), T-cell attracting chemokines (i.e., CCL3, CXCL9), generate oxidative burst, they can also activate dendritic cells. N1 is characterized by increased adhesion, transmigration, phagocytosis, and NET release capacity [56,57]. N2 neutrophils have pro-tumor and anti-inflammatory functions and secrete IL-10 and TGF-β1 [56,57]. This classification is similar to the classification of tumor-associated macrophages (TAM), respectively, M1 and M2 [56,57]. The other subpopulation of neutrophils is rTEM, named after the process of reverse transendothelial transmigration. This mechanism is associated with the adhesion molecule JAM-C at the junction of vascular endothelial cells [58]. An elevated number of rTEM neutrophils was observed while JAM-C was downregulated [59]. rTEM neutrophils have a different surface receptor repertoire than other neutrophils (CD54 high, CXCR1 low). These neutrophils had increased expression of adhesion molecule-1 (ICAM-1; CD54) and decreased expression of the chemokine receptors CXCR1 and CXCR2 [11]. The study reveals that reverse-transmigrated neutrophils produced increased amounts of ROS and reduced levels of apoptosis. Elevated number of rTEM neutrophils was observed in patients with chronic inflammation in comparison to the control group [11]. Therefore, it is important to understand and control the reverse migration of neutrophils and prevent the inflammation from transforming into a chronic one.

Neutrophils are important immune cells for the early and relapse phase of MS [16,22,23]. They impact the permeability of BBB [6,37,38,39,57], create a pro-inflammatory environment by cytokine secretion [19,32,33,35,38,39], regulate the antigen presentation process [7,54], and correlate with MS severity. A promising indicator, in the context of neutrophils, is the NLR, which is elevated in MS patients in general, related to disease activity and disability status (Figure 1). In the future, it may facilitate the diagnosis and early treatment of patients with MS [16,20,21,22,23,24,25,26].

### 2.5. Neutrophils and DTM Drugs

Over the past decade, patients with MS have access to an increasing number of disease-modifying therapies (DMTs). All of these drugs interfere with the immune system of patients despite their mechanism of action [60]. In one study, it was shown in in vitro models that most common DMTs (i.e., high-efficient, second-line: fingolimod (FTY) and natalizumab (NAT) and moderate-efficient injective, first-line: interferonβ-1b (IFNβ-1b), glatiramer acetate (GA)) affect neutrophil functions. All of these DMTs impaired the ability of neutrophils to kill Klebsiella pneumoniae [61]. Only FTY exposure negatively influenced Klebsiella pneumoniae-induced production of ROS in PMNs [61]. In another study, a general increased risk of infection was observed among patients with MS taking DTMs (rituximab (RTX), NAT, FTY, IFNβ, GA) [60]. The lowest rate of infection was observed with IFNβ and GA, and the highest rate of serious infections was associated with taking RTX [61]. Torgauten et al. observed neutropenia during RTX therapy in patients with RRMS, SPMS, and PPMS [62]. In contrast, neutropenia was not observed during ofatumumab treatment in RRMS [63]. Moreover, if neutrophil levels were lowered by teriflunomide, they returned to baseline levels after the switch to ofatumumab [63]. The risk-benefit assessment of DMTs should be considered when choosing MS therapies [60]. Taken together, the most significant side-effect of MS therapy lowering the neutrophil ratio is serious infections.

Considering the available data, the role of neutrophils in the pathogenesis of MS is very important, most important during the development of the disease and during its relapse. Neutrophils act in many directions in the pathogenesis of MS. The short lifespan of neutrophils may be the cause of underscoring their role in demyelination and in the pathogenesis of MS [3]. Neutrophils are functionally important for the development of CNS inflammation and initiation of MS and EAE, which depend on their infiltration into the CNS and secretion of soluble factors that contribute to the maturation of professional APCs during CNS inflammation [43]. Their role in this disease is noticeable but still not fully unraveled and remains a topic for further research.

## 3. The Role of Neutrophils in Stroke

Stroke is one of the leading causes of lifetime morbidity and a major cause of death in industrialized countries [64]. The inflammatory response in stroke leads to oxidative burst with ROS release and promotes an immunological response associated with cytotoxicity and brain damage [65,66]. Elements of the innate immune system, e.g., cytokines, chemokines, or specialized cells, generated peripherally or locally in the brain, actively participate in processes implicated in tissue damage [66,67,68]. The major subtypes of stroke are ischemic and hemorrhagic stroke, representing approximately 80% and 20% of cases, respectively [69,70]. The most common, brain ischemia, is caused by the occlusion of a blood vessel, resulting in the sudden disruption of blood flow in a region of the brain [71]. The obstruction of the blood flow by a thrombus leads to cerebral injury and subsequent disability [72].

Rupture of atherosclerotic plaque and subsequent atherothrombosis are the pathological hallmarks of acute ischemic stroke (AIS) [73]. AIS is the main cause of adult permanent disability, the second most common cause of dementia, and the third most common cause of mortality worldwide [74]. AIS is a hypoxic-ischemic disorder associated with a sterile inflammatory reaction [75]. This encompasses both an acute and a prolonged inflammatory process, which is characterized by rapid activation of microglia, production of proinflammatory molecules, and early infiltration of neutrophils into the affected brain region, followed by the migration of monocytes and T cells into the brain parenchyma [76].

In normal brain tissue, inflammatory mediators are present at low levels. However, during ischemic conditions, they are elevated and result in immune cell recruitment, which represents an important mechanism of the progression of brain lesions [77]. Neutrophils are the first white blood cells (WBC) recruited into the damaged territory of the brain within the first hours after stroke onset [78,79]. The role of neutrophils in the pathogenesis of stroke has gained more attention recently. They exert direct neurotoxic functions through releasing proteolytic enzymes or indirect as a result of their intravascular accumulation, capillary blood flow obstruction, and the no-reflow phenomenon [80,81,82,83]. The release of various mediators from these cells results in additional tissue damage and poor neurological improvement [84,85]. Their counts are elevated a few hours after stroke and remain high for a week [85]. Neutrophils release proteases, like elastase and proteinase 3, and peptidases, like MMP-9, which disrupt the BBB. They also release various cytokines that attract other immune cells to the site of inflammation [86,87] (Figure 2). Increasing data have shown that neutrophils cross-talk, shape the maturation and effector functions of other leukocytes, show diverse phenotypes, and play important roles in several pathological processes [88,89,90,91,92]. It was speculated that neutrophils play a more important role in ischemia/reperfusion than in permanent stroke [93,94]. This assumption is based on the observation that, in ischemic stroke, blood flow is arrested, and peripheral leukocytes cannot migrate into the ischemic region through the circulation unless they reach this territory through the collateral blood vessels, peripheral regions or after reperfusion has been present [82,95]. It has been observed in the rodent Middle Cerebral Artery Occlusion model (MCAO) that the number of neutrophils increased with time in the ischemic region [9,79]. Authors have observed that a major pathway of neutrophil invasion into the ischemic tissue during the arterial occlusion was via leaving the circulation at the leptomeningeal vessels [96]. The presence of neutrophils in the leptomeninges surrounding the infarcted region was also confirmed in human brains post-mortem, suggesting that this pathway may also be relevant in human stroke [96]. 

A recent study by Datsi et al. gave a new perspective on neutrophil role in stroke pathogenesis. Their dynamic studies indicated that neutrophils obtained from patients were more prone to form NETs than from controls. Moreover, these NETs were more difficult to lyse by DNAse-I treatment. Authors have suggested that neutrophil dysfunction was present before the onset of diseases and plays a key role in stroke pathogenesis rather than being the effect of its onset [97]. Elevated generation of NETs by stroke patients’ neutrophils was also observed by Gao et al. [98]. This result emphasizes the need for more detailed studies on neutrophil biology and function in various pathological states.

### 3.1. Phenotype Plasticity of Neutrophils and Their Role in Stroke

Neutrophils have long been considered a highly conserved cell population. Nowadays, there is increasing evidence that neutrophils have the ability to switch towards various phenotypes depending on specific microenvironments [9,90,99]. The inflammatory process can increase their lifespan. This process, called priming, expands the half-life of neutrophils in response to cytokine exposure. Other subpopulations of neutrophils have the ability to travel back to the bloodstream after infiltrating the injured tissue. They are called rTEM [10]. There is also a subtype called N2, or immunosuppressive, which contributes to the resolution of inflammation and promotes clearance of neutrophils by macrophages or microglia [9,87]. The results strongly suggested that altered homeostasis of neutrophil subpopulations, together with their hyperactivation, is correlated with the clinical severity of acute stroke. Weisenburger-Lile et al. have shown in their studies that there was a basal hyperactivation of circulating neutrophils during acute ischemic stroke [100]. Authors have characterized this state of neutrophil activation by the lower L-selectin expression and higher expression of CD11b on their surface and elevated ROS and elastase production. Such hyperactivation was associated with an imbalance between apoptotic and necrotic neutrophils [100]. There was also deregulation of various subsets of these cells. It has been shown that in patients, the percentage of senescent neutrophils and those with a reverse transendothelial migration capacity were elevated compared to control subjects [100]. The elevated level of rTEM neutrophils was associated with an increased level of sJAM-C—a soluble form of adhesion molecule important for neutrophil transmigration [100,101]. This subset is also characterized by prolonged survival and enhanced oxidative burst [11]. This might explain, at least in part, the higher ROS production in stroke patients and the lower rate of neutrophil apoptosis [100]. Physiologically, neutrophils constitutively undergo apoptosis, which is important for the resolution of inflammation. The necrotic process acts in the opposite way, inducing the release of cytotoxic factors that can exacerbate tissue damage [102]. Such alterations were associated with the clinical severity of stroke, according to the NIH Stroke Scale score [100]. Contrary to these subsets, N2 or immunosuppressive neutrophils exhibit reduced proinflammatory features (like decreased adhesion properties and less ROS production) [103,104]. Cai et al. have observed in the tMCAO model that the expression level of the N2 subtype marker—CD206 was relatively stable over time in neutrophils infiltrating into the brain lesion [9]. Authors have suggested that N2 neutrophils facilitated neutrophil clearance by macrophages, which promotes neural inflammatory resolution and does not further induce neuronal death. Using the tMCAO model, it was shown that skewing neutrophils toward the N2 phenotype before stroke induction resulted in a reduced infarct volume [9]. These results suggested that the frequency of neutrophil subtypes may have an important impact on stroke outcomes. Moreover, there are suggestions that various phenotypes of neutrophils may have various potentials to form NETs [105]. However, this issue and its implications for stroke pathogenesis need more in vitro and in vivo studies.

### 3.2. The Role of NETs in Thrombosis

In response to various stimuli, like microorganisms, antibodies, activated platelets, and reactive oxygen species, neutrophils may release NETs, which are large web-like structures composed of DNA, histones, and granule proteins, such as elastase, and MPO [106,107]. Among the many inflammatory cells and factors promoting thrombosis, there is increasing evidence that neutrophils and their NETs play an important role in this process [108,109]. It was observed that in patients with AIS, NETs are present in blood clots or plasma [110,111,112]. NET structures contain tissue factor (TF), von Willebrand Factor (vWF), and histones, acting as a scaffold for platelet adhesion, activation, and aggregation [110]. Platelet-neutrophil crosstalk resulting in the NET formation is recognized as one of the main causes of inflammation and thrombosis. It seems that cell activation is bidirectional [113]. Activated platelets may stimulate neutrophils to release NETs via P-selectin glycoprotein ligand-1 (PSGL-1) interactions [114]. P-selectin is one of the molecules from platelets alpha-granules, which is an important adhesion molecule that binds to PSGL-1 present in immune cells [115,116]. In mouse permanent MCAO model (pMCAO) studies, it was observed that during the early stages of inflammation, neutrophils recruited to the injured vessels use their PSGL-1 domain to scan for the presence of activated platelets [116,117]. Another mechanism for platelet stimulation of NET formation is based on the high-mobility group box 1 (HMGB1) expression on platelet-derived microparticles (PMPs) [98]. It was shown that the expression of HMGB1 was upregulated on PMPs obtained from AIS patients, and it induced NET formation through the autophagy process [98]. Additionally, neutrophil-derived NETs can activate platelets [118]. Phosphatidylserine-bearing (PS-bearing) NETs provide functional platforms for PMPs and coagulation factor deposition and thus increase thrombin and fibrin formation [118]. Moreover, it was reported that the release of extracellular DNA during NETosis and PS exposure was associated with elevated fibrin deposition and the formation of prothrombinase complex [118]. Thus, NETs with functional PS are able to induce thrombin generation and platelet activation, leading to an increased risk of atherothrombosis [118]. Indeed, elevated levels of NET markers were reported in the systemic blood of AIS patients [98,112]. Recent studies have suggested that NET content in thrombus is associated with poor outcomes in stroke patients and may be responsible for reperfusion resistance [111,119,120,121]. Studies by Peña-Martínez et al. utilizing two in vivo models of platelet-rich thrombosis observed that inhibition of NETs formation results in the hamper of the formation of stable clots and did not cause any ischemic lesions. Overall, these results suggest that the development of platelet-rich thrombus depends on NET formation [122]. Moreover, recent studies observed differences in neutrophil presence and NET expression in clots from various etiologies, indicating that this may add information to the mechanism of clot formation [111,123,124,125]. It was observed that cardioembolic clots have a higher content of NETs in comparison to both atherothrombotic and cryptogenic clots [111,123,124,125]. Additionally, Jabrah et al. have shown differences in NET distribution in clots of various etiology, located either at the periphery of the thrombus, in the center, or a combination of both [123].

### 3.3. The Impact of Neutrophils on BBB Function

The BBB forms a protective barrier, limiting the transport of various factors into the CNS and thus maintaining brain homeostasis [126]. During ischemia, the integrity of BBB is disrupted, allowing the transmigration of immune cells into the brain. It is well known that neutrophils contribute to the BBB loss of integrity [127,128]. ROS release from neutrophils promotes MMP activation, which disrupts the basement membrane of the neurovascular unit [129,130]. One of the crucial MMPs is MMP-9. Neutrophils are the major source of this MMP after ischemic stroke. MMP-9 has been shown to be important in postischemic BBB disruption, immune cell infiltration, and subsequent brain damage [131,132]. Moreover, senescent neutrophils have been shown to rapidly migrate to the inflammatory site in the in vivo murine model, and this might also be true in patients with ischemic stroke [133]. The release of neutrophil elastase might also induce vascular damage, and together with the altered equilibrium between apoptotic and necrotic neutrophils, these may aggravate ischemia and acute inflammation [81,100]. Another factor contributing to BBB damage is NETs, releasing many cytotoxic proteases like histone protease, elastase, and MPO, which directly induce endothelial cell damage, resulting in elevated permeability of BBB [134,135]. Kang et al. have also shown that increased NET formation or altered NET clearance was detrimental to revascularization and vascular repair after stroke [136]. Additionally, they showed in a murine stroke model that neutrophil depletion results in reduced BBB impairment 14 days after stroke. Moreover, it was observed that such depletion increased neovascularization and vascular perfusion, pointing to the important role of neutrophils not only in BBB damage but also in repair processes [136].

### 3.4. NLR as Diagnostic and Prognostic Tool for Stroke

Inflammation plays an important role in the initiation and progression of atherosclerosis [137]. WBC count and count of their subtypes are commonly used and easily obtainable inflammatory markers [138]. The neutrophil-to-lymphocyte ratio (NLR) has been used as a marker of, for example, cardiovascular diseases, cancer, and diabetes [139,140,141]. Nowadays, there are numerous studies investigating the relationship between the NLR ratio and various stroke outcomes [142,143,144]. Wang et al. observed that higher NLR was positively associated with the risk of hemorrhagic transformation (HT) and 3-month death after stroke [143]. Moreover, several studies have indicated that baseline NLR level is higher in patients with ischemic stroke than with hemorrhagic stroke [145,146]. It was indicated that NLR is a predictor for stroke progression and unfavorable functional outcomes [147]. The prognostic value of NLR is altered by several factors. The time point after stroke is one of them. Wang et al. have shown that NLR at 48–72 h post-stroke was a predictor of an unfavorable outcome [143]. Studies conducted by Tokgoz et al. have indicated that NLR assessed at the time of hospital admission may be a predictor of short-term mortality. This was independent of infarct volume in AIS patients [144]. It has also been suggested that higher NLR may estimate the size of the infarct independently from the etiology [148]. There are studies correlating NLR not only with stroke type but also with stroke territory. It has been suggested that NLR correlated with infarct volume in anterior circulating stroke (ACS) but not with posterior circulating stroke (PCS) infarct volume in AIS. NLR is also an independent predictor of 3-month mortality in AIS patients [149]. It was also shown that higher NLR can indicate stroke severity on admission and help to make a decision on the necessity of endovascular intervention, especially in ACS [149]. Neutrophil count and NLR may also predict parenchymal hemorrhage and 3-month poor outcome in AIS patients after pharmacological thrombolysis [150]. Furthermore, elevated NLR levels had detrimental effects on prognosis due to secondary brain injury by neutrophil activation and increased risk of infection by lymphocyte suppression [151]. The mechanism underlying the clinical significance of NLR in stroke is related to the central role of inflammation in all subtypes of this disease and the importance of this process in all stroke phases—from the initiation to the recovery [152,153,154,155,156,157]. The NLR is a promising biomarker for stroke patients, as it is low-cost and easy to obtain during standard laboratory procedures. More research regarding the dynamics and temporal changes of NLR may benefit the prognostication and stratification of stroke patients. Moreover, there is a great need for optimization of NLR ranges for patients with co-existing diseases, like hyperglycemia or diabetes [155,156,157].

### 3.5. The Role of Neutrophils in Reperfusion Process

As ischemic cerebrovascular diseases are one of the important causes of mortality and disability, reperfusion of an ischemic brain is an important process for reversing brain damage after stroke [158]. Alterations in neovascularization after brain injury worsen outcomes after cerebral ischemia [159,160]. However, reperfusion may lead to secondary brain damage induced by the inflammatory reaction, which is called ischemia/reperfusion (I/R) injury [161,162]. There are only two therapies approved so far for AIS—pharmacological thrombolysis via recombinant tissue plasminogen activator (r-tPa) leading to degradation of fibrin in the thrombus and mechanical removal of the thrombus [163]. Several clinical trials have proved the efficacy and benefits of treatment with r-tPa in AIS patients within 4.5 h from symptom onset [164,165]. It is often the first choice therapy within this time window. However, about half of patients are still disabled or even die. Several factors related to the therapy outcome have been identified. However, the detection of novel factors is still enormously important [166]. It has been shown that the rate of early recanalization is rather low (less than 50%), especially in the case of platelet-rich thrombi, which are resistant to thrombolytic drugs because they contain little or no fibrin [167,168,169]. The therapy with tPa is also related to an increased risk of intracerebral hemorrhage, and factors contributing to hemorrhage are still not fully unraveled [170,171,172]. tPa can activate the brain endothelium, leading to the degradation of vascular integrity and BBB breakdown [173,174]. It was shown that neutrophils, as one of the first cells migrating to the site of damage, participate in BBB damage by the release of cytotoxic proteases and proinflammatory mediators [96]. It has been shown that tPa promotes the recruitment of neutrophils to the ischemic tissue [175,176]. Moreover, it has been suggested that the higher level of neutrophils and NLR on admission is associated with an increased risk of HT and a 3-month poor outcome after r-tPa treatment [177,178]. However, results from Ying et al. indicate that elevated neutrophils and NLR 24 h after r-tPa therapy, but not that on admission, are associated with PH and 3-month poor functional outcomes [150]. Activated neutrophils release NETs that are considered a major trigger of inflammation and thrombosis [179,180]. The work by Wang et al. suggested that tPa directly induces NET formation under ischemic conditions [181]. NETs present in the thrombi were shown to complicate their dissolution and removal, resulting in worse clinical outcomes after recanalization therapies [182,183,184,185]. Abbasi et al. showed that older thrombi (>4 h post-stroke onset) showed elevated levels of citrullinated histone H3 (the marker of NETs) and were stiffer, which makes them harder to dissolve by r-tPa [165]. Novotny et al. have confirmed this observation in their study [121]. It was observed that targeting NETs by DNase-I recanalizes occluded vessels, improving stroke outcomes [122]. There are assumptions from in vivo studies that degradation of NETs or inhibition of its formation significantly enhanced neovascularization, pointing to a new potential therapeutic target [136]. 

Although there were promising results from animal models related to the therapeutical effect of neutrophil depletion before or at the onset of stroke, human trials did not confirm these observations [10,186,187]. Moreover, several clinical trials have shown an increase in adverse events, including increased incidence of infection and hemorrhagic transformation, and also increased stroke mortality [188]. Administration of CD11/CD18 antagonists in patients receiving r-tPa also did not improve stroke outcomes [10]. The same was observed for natalizumab—monoclonal antibody targeting VLA-4 and for humanized monoclonal antibody against CD11/CD18 [189,190,191]. This may be due to, among others, the fact that activated neutrophils exhibit accelerated phagocytic activity, clearing cellular debris during AIS, which is a two-faced process. It may exacerbate inflammation but also promote the repair process [192]. They are also important for angiogenesis, stimulating this process by secretion of VEGF [193]. Moreover, the functional differentiation of neutrophils may be responsible for these results. Indeed, studies conducted on the MCAO model indicated that increased infiltration of N2 neutrophils into the CNS results in a reduction of infarct volume [88,194]. Thus, additional studies must be carried out to explain the functions of different subsets of neutrophils and their importance in various phases of stroke.

## 4. Discussion

Inflammatory reaction is an important contributor to the pathology of MS and stroke. Inflammatory mediators encompass reactive oxygen species, cytokines, chemokines, and elements of the complement system [1]. They promote immune cell recruitment into the damaged area of the brain and activation of resident and peripheral immune cells, which promotes cytotoxicity and additional brain damage [1,2]. One of the cell types present in the brain at an early phase of inflammation is neutrophils. The role of neutrophils in CNS disorders, however, is complex. Moreover, increasing evidence has shown that neutrophils possess the ability to switch toward various phenotypes depending on specific microenvironments [9,90,99,195]. It is important to assess the secretory potential of various neutrophil subtypes and their ability to form NETs, as they play an important role in various aspects of MS and stroke pathology [104,196]. Thus, more studies are needed regarding these subtypes and their function in various CNS disorders and stages of disease development. 

The NLR is a promising biomarker for numerous diseases, like infection diseases, sepsis, and major cardiac events, and also for both MS and stroke, as it reflects severe inflammation and a compromised immune system. [16,151,197,198,199]. It is a low-cost and easily obtainable blood-based marker. Numerous studies have proved its utility for various pathological aspects of CNS disorders [22,23,24,142,143,144]. However, there are ambiguous results regarding its optimal threshold and the most informative time point of analysis. Thus, more studies have to be conducted to consider the dynamic changes of NLR. Moreover, an analysis of the impact of co-existing diseases on NLR ranges should also be carried out.

Studies conducted on NLR’s potential as a prognostic marker, both for MS and stroke patients, pointed to the important role not only for the inflammatory process but also for the neutrophils. Indeed, despite various etiologies of both diseases, the role and importance of neutrophils in their development and progressions reveal many similarities. In both diseases, neutrophils are one of the first cell types migrating into the affected region, which makes them important players in the early phase of MS and stroke [17,18,27,78,79]. Their secretory activity leads to exacerbated inflammatory processes and BBB alterations and induces the migration of other peripheral immune cells into the CNS, aggravating the pathological processes [28,32,37,84,85,86]. The ability of neutrophils to NET formation is also an important aspect in the pathogenesis of both MS and stroke. However, the detailed role of NET varies between these disorders [32,108,156]. The effects of phagocytic activity of neutrophils are similar, as it may be detrimental, leading to damage of CNS-resident cells, but on the other hand, it leads to cellular debris clearance, which supports repair processes [40,193]. The impact of various subtypes of neutrophils is weakly recognized as far. However, the results suggested their important role in both diseases [9,11,100]. The role of neutrophils in neuron damage is more pronounced in multiple sclerosis. Uncontrolled action of these cells results in demyelination and exacerbation of inflammatory and neurodegenerative processes [3,41,200]. Moreover, their role as alternative APCs is also important for this disease [6,7]. Recent results suggest that alterations in neutrophil function precede stroke onset rather than just being the effect of disease development, which may be the major difference in the role of neutrophils in the pathology of MS and stroke [97,98]. Moreover, the role of neutrophils and NETs in clot formation and ischemia/reperfusion injury made them one of the main factors contributing to stroke development and its exacerbation [93,94,182,183,184,185] (Figure 3).

Therapy regarding neutrophil depletion or inhibition has not succeeded. Rather than global action on the neutrophil population, targeted approaches may be more successful. Exploratory clinical trials inhibiting MMP-9, factors expressed mainly by neutrophils, have gained some promising results, demonstrating reduced alteration of BBB and neural tissue damage, both processes important in MS and stroke pathology [201]. Another potential therapeutic target is NET, as its importance for the pathology and progression of both diseases was shown [19,32,97,202]. Procedures aimed at regulating the number and functions of neutrophil subpopulations may be beneficial for numerous disorders. The N2 subtype, which is characterized by anti-inflammatory features, may be crucial for the resolution of inflammation and induction of repair processes [52,53,79,91]. Platelet-neutrophil interaction is a promising target for stroke therapy, as it results in the formation of NETs and clots more resistant to revascularization therapy [108,109,179]. Regulation of neutrophil migration and its secretory activity may be beneficial for stroke and MS patients. However, current data (e.g., the effectiveness of natalizumab in MS, but not in stroke) suggested that these processes need more detailed studies, and it should be emphasized that adequate time-point for such interventions is very important, as it may lead to diverse effects.

## 5. Conclusions

Neuroimmunology and neuroinflammation have become fields of increased interest in searching for novel therapeutic targets for CNS disorders. Previous experience in therapeutical approaches has pointed to the importance of specific targeting of selected molecules rather than action on, e.g., whole cell population. Results presented in this review have confirmed this point of view in the context of neutrophils. Their role in the pathogenesis of multiple sclerosis and stroke is complex and ambiguous and needs more in-depth studies. Therapeutic intervention based on neutrophil subtypes or specific molecules expressed by them should also take into consideration adequate time for their implementation. 

## Figures and Tables

**Figure 1 brainsci-14-00423-f001:**
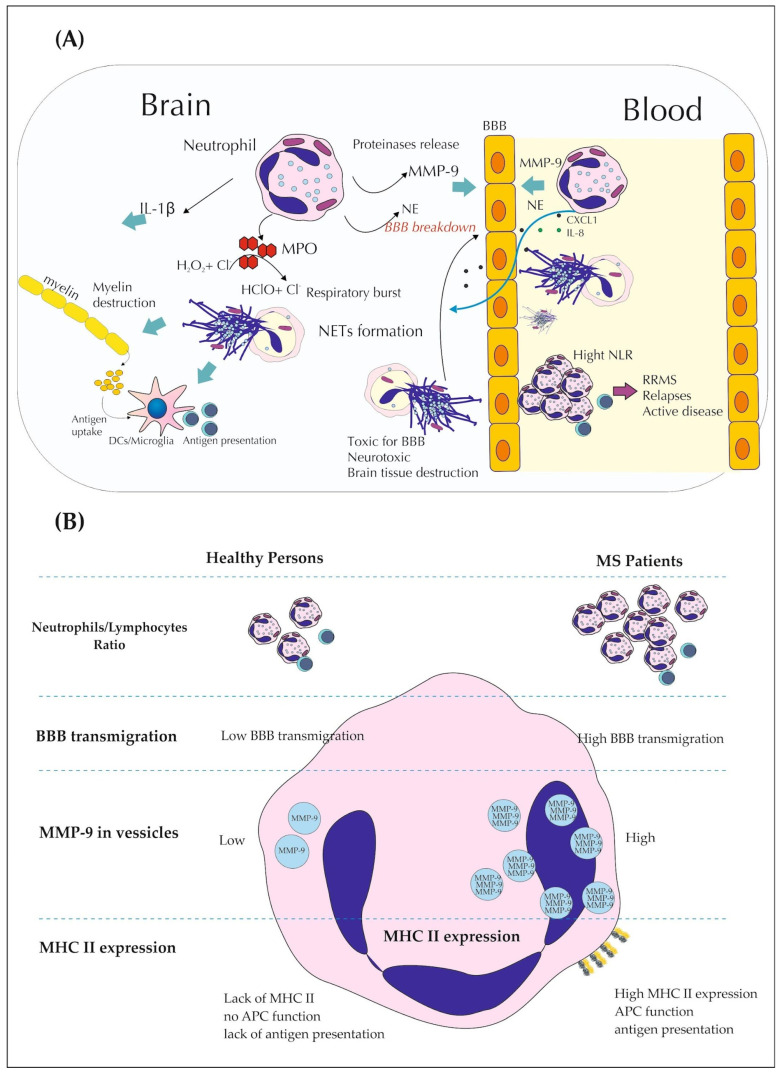
The role of neutrophils in multiple sclerosis pathogenesis. (**A**)—An overview of the function of neutrophils in multiple sclerosis. (**B**)—Comparison of the neutrophils’ action in healthy persons and MS patients. MS—multiple sclerosis, BBB—blood–brain barrier, MMP-9—matrix metalloproteinase 9, NE—neutrophil elastase, MPO—myeloperoxidase, NETs—neutrophil extracellular traps, DCs—dendritic cells, NLR—neutrophils-to-lymphocytes ratio, RRMS—relapsing-remitting MS, MHC—major histocompatibility complex, APC—antigen-presenting cells.

**Figure 2 brainsci-14-00423-f002:**
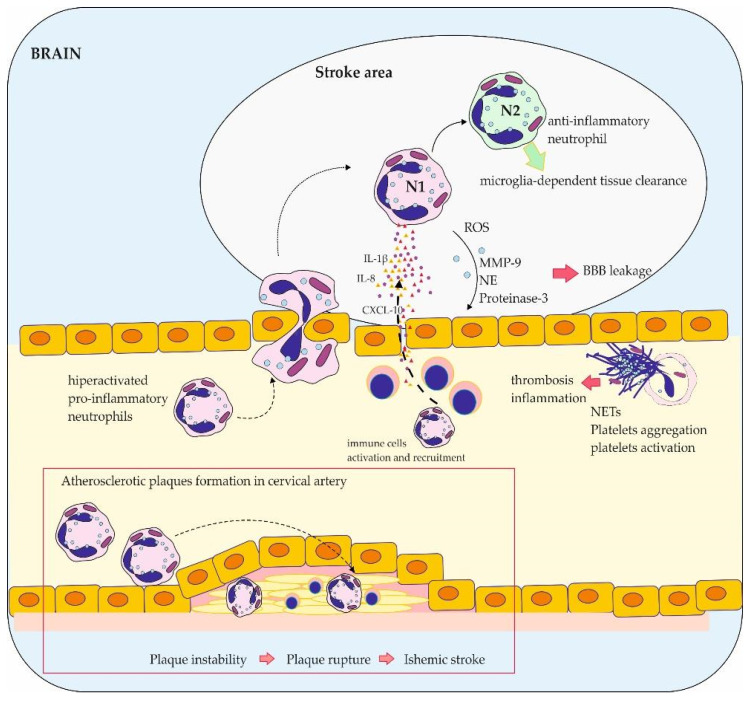
The role of neutrophils in stroke pathogenesis and thrombosis. ROS—reactive oxygen species, MMP-9—matrix metalloproteinase 9, NE—neutrophil elastase, BBB—blood–brain barrier, NETs—neutrophil extracellular traps.

**Figure 3 brainsci-14-00423-f003:**
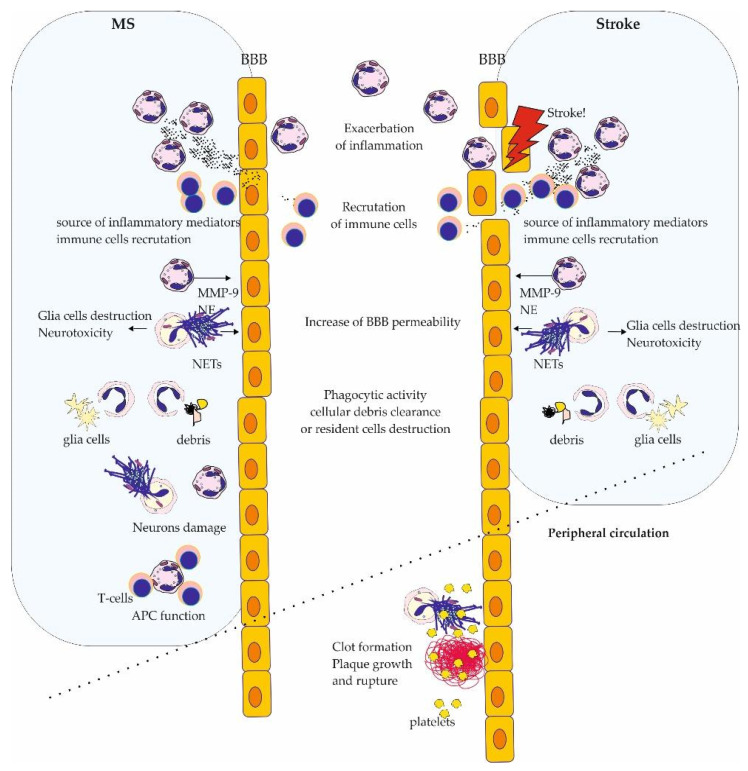
The similarities and differences in neutrophils function in pathogenesis of multiple sclerosis and stroke. MS—multiple sclerosis, BBB—blood–brain barrier, MMP-9—matrix metalloproteinase 9, NE—neutrophil elastase, NETs—neutrophil extracellular traps, APC—antigen-presenting cells.

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
