# Peer review of "The Role of Neutrophils in Multiple Sclerosis and Ischemic Stroke"

_brainsci, 2024, doi:10.3390/brainsci14050423_

Round 1

Reviewer 1 Report

Comments and Suggestions for Authors

In this review the authors discussed the role of neutrophils in multiple sclerosis (MS) and ischemic stroke. Some concerns and suggestions are listed as below:

The role of neutrophils in different disease stages of MS should be discussed, for example, CIS VS. RRMS VS. SPMS. Did you observe sex differences?

Regarding NLR in MS, some may argue that this is not MS-specific since elevated NLR can also be noted in other neurological diseases.

Related findings (neutrophils in the pathogenesis of MS and stroke) in this review should be summarized using figures or tables.

Related findings from clinical studies and preclinical studies should be discussed separately. Potential differences should also be discussed.

The effects of DMT drugs on neutrophils and MS should not be ignored.

How should we target neurrophils for treating MS and stroke?

Any potential side effects following neutrophil depletion or the inhibition of their function?

In lines 235-236, please avoid using 'seems to be very important'.

Studies regarding neutrophils at the single-cell level should be discussed.

How can we measure functional changes of neutrophils in patients with MS and stroke?

Any relationship between neutrophils and ARR/EDSS?

Comments on the Quality of English Language

fine

Author Response

In this review the authors discussed the role of neutrophils in multiple sclerosis (MS) and ischemic
stroke. Some concerns and suggestions are listed as below:
The role of neutrophils in different disease stages of MS should be discussed, for example, CIS VS.
RRMS VS. SPMS. Did you observe sex differences?
Thank you for this suggestion. We have added more information regarding the relationship between
NLR and various MS stages, and also impact of the sex.
Regarding NLR in MS, some may argue that this is not MS-specific since elevated NLR can also be
noted in other neurological diseases.
We haven’t stated that NLR is MS-specific. However, as it may be misleading for readers, we
emphasized this issue in MS and stroke sections, and also in discussion.
Related findings (neutrophils in the pathogenesis of MS and stroke) in this review should be
summarized using figures or tables.
Thank you for your suggestion. We have added figure summarizing similarities and differences in
neutrophil role in stroke and MS
Related findings from clinical studies and preclinical studies should be discussed separately. Potential
differences should also be discussed.
We have mentioned similarities and differences in the role of neutrophils in both diseases in the
discussion section. We haven’t discuss this issue in details, as our review is already extensive in
volume, and all information are already mentioned in the sections regarding multiple sclerosis and
stroke.
The effects of DMT drugs on neutrophils and MS should not be ignored.
Thank you for your suggestion. We have described the current knowledge about the relationship
between DMT drugs and neutrophils in MS.
How should we target neurrophils for treating MS and stroke?
According to your suggestion, we have added information about potential targeting of neutrophils for
treating MS and stroke in the discussion section.
Any potential side effects following neutrophil depletion or the inhibition of their function?

Potential side effects have been described in the multiple sclerosis and stroke sections (line 289,297
and 551-553, respectively).
In lines 235-236, please avoid using 'seems to be very important'.
This sentence has been modified.
Studies regarding neutrophils at the single-cell level should be discussed.
Thank you for your suggestion. We agree that analysis of the results from the single-cell level are
important, however we decided to omit this field, as our paper was already extensive, discussing the
complex role of neutrophils and their function in two diseases with different etiological background.
How can we measure functional changes of neutrophils in patients with MS and stroke?
Most articles cited by us have measured functional changes of neutrophils by the analysis of
alterations in their secretory activity, and ex vivo studying their ability to, e.g., NETs formation,
migration and BBB disruption. Functional changes may be also analysed utilizing transcriptomic and
proteomic analyses.
Any relationship between neutrophils and ARR/EDSS?
We have added information about the relationship between neutrophils and ARR/EDSS.

Reviewer 2 Report

Comments and Suggestions for Authors

The authors wrote a review about the role of neutrophils in multiple sclerosis and ischemic stroke.

However, it is not clear at all why they have chosen only these two disorders.

There is no clear aim.

The big lack of this review is no figures, schemes or similar.

The references need to be updated.

Comments on the Quality of English Language

needs grammar check

Author Response

Thank you for you remark. We have cleared the issue, why we choose these two disorders as a topic
of our review.
There is no clear aim.
We have clarified the aim of our article.
The big lack of this review is no figures, schemes or similar.
Thank you for your suggestion. We added three figures.
The references need to be updated.
The references have been updated.

Reviewer 3 Report

Comments and Suggestions for Authors

The topic is relevant due to the increasing incidence of stroke and MS among young people with increasing mortality and disability from these diseases. The response of neutrophils to vascular and autoimmune diseases is not well understood. This fact requires additional research that can help in the prevention and treatment of these diseases.

The list of authors is not specified according to the journal rules.

Almost the majority of the introduction is devoted to known information that can be found in an any internal medicine textbook. Please add additional information on your topic from recent research studies to highlight the need for this review.

The purpose of the study should be more specific.

You decided to discuss the role of neutrophils in neuroinflammatory processes in two different diseases in etiology and pathogenesis. In my opinion, it is better to talk about each of these diseases separately. Or briefly explain in the introduction why you chose these diseases in your review.

Much information requires reference links. Add reference to lines 72, 73, 77, 153, 155, 200, 202, 204, 209, 214, 215, 223, 229, 231, 234, 243, 252, 281, 359, 361,

The -/-  signs in the line 160 are unclear

There is no discussion in your review, but the conclusion is stated in many details. You can add additional information to the conclusion and label it as a discussion, as well as add a short conclusion with practical recommendations.

References are not formatted by journal roles. Many references are over 15 years old.

Author Response

The topic is relevant due to the increasing incidence of stroke and MS among young people with
increasing mortality and disability from these diseases. The response of neutrophils to vascular and
autoimmune diseases is not well understood. This fact requires additional research that can help in the
prevention and treatment of these diseases.
The list of authors is not specified according to the journal rules.
The list of authors have been corrected.

Almost the majority of the introduction is devoted to known information that can be found in an any
internal medicine textbook. Please add additional information on your topic from recent research
studies to highlight the need for this review.
We have modified the introduction section according to your suggestion.
The purpose of the study should be more specific.
We have specified the aim of our review.
You decided to discuss the role of neutrophils in neuroinflammatory processes in two different
diseases in etiology and pathogenesis. In my opinion, it is better to talk about each of these diseases
separately. Or briefly explain in the introduction why you chose these diseases in your review.
We have explained the reason of choosing multiple sclerosis and stroke as subjects of our study.
Much information requires reference links. Add reference to lines 72, 73, 77, 153, 155, 200, 202, 204,
209, 214, 215, 223, 229, 231, 234, 243, 252, 281, 359, 361,
The references have been added.
The -/-  signs in the line 160 are unclear
The sign has been corrected.
There is no discussion in your review, but the conclusion is stated in many details. You can add
additional information to the conclusion and label it as a discussion, as well as add a short conclusion
with practical recommendations.
Thank you for your suggestion. We have modified the conclusion section and renamed it as a
discussion, and add conclusions separately.
References are not formatted by journal roles. Many references are over 15 years old.
The references have been updated, and the format has been changed.

Round 2

Reviewer 1 Report

Comments and Suggestions for Authors

The authors have addressed previous concerns.

Comments on the Quality of English Language

fine

Author Response

Thank you for your opinion.

Reviewer 2 Report

Comments and Suggestions for Authors

The authors have made progress with their review.

One minor comment for the abstract:

Here should be changed the aim to be clearer in order to understand why the authors have chosen these two CNS disorders as well as they did in the introduction.

Author Response

We modified the abstract to clarify the aim of our article.

Reviewer 3 Report

Comments and Suggestions for Authors

The revised version, in my opinion, has become more interesting, scientific and informative for readers.

The topic is very relevant and corresponds to the content.

The purpose of the study is stated clearly and specifically.

Materials and methods are described in detail and clearly.

The results are meaningful and explained by illustrations. The authors used successful statistical analysis to prove the significance of their results.

The discussion is organized logically based on the results obtained in comparison with other studies from the literature.

The conclusion is specific and specific, with ideas for recommendations for future research on the topic.

Please continue to update the references. Please note: the journal number is in italics, the year of publication is in italics and bold.

Author Response

The references have been updated, and the format of references has been changed according to instructions for authors available on website of Brain Sciences Journal, in which the journal number is in italics, the year of publication is in bold.